# Immunotherapy against Systemic Fungal Infections Based on Monoclonal Antibodies

**DOI:** 10.3390/jof6010031

**Published:** 2020-02-29

**Authors:** Camila Boniche, Suélen Andreia Rossi, Brenda Kischkel, Filipe Vieira Barbalho, Ágata Nogueira D’Aurea Moura, Joshua D. Nosanchuk, Luiz R. Travassos, Carlos Pelleschi Taborda

**Affiliations:** 1Biomedical Sciences Institute, Department of Microbiology, University of São Paulo, Sao Paulo 05508-000, Brazil; camila.boniche@usp.br (C.B.); su.arossi@gmail.com (S.A.R.); brendakischkel@gmail.com (B.K.); flpxbarbalho@gmail.com (F.V.B.); 2Tropical Medicine Institute, Department of Dermatology, Faculty of Medicine, University of Sao Paulo, Sao Paulo 05403-000, Brazil; agatandmoura@gmail.com; 3Departments of Medicine (Division of Infectious Diseases) and Microbiology and Immunology, Albert Einstein College of Medicine, New York, NY 10461, USA; nosanchuk@gmail.com; 4Department of Microbiology, Immunology and Parasitology, Federal University of São Paulo, Sao Paulo 04021-001, Brazil; luiztravassos@gmail.com

**Keywords:** therapeutic vaccines, monoclonal antibodies, systemic fungal infections, immunotherapy, antifungal vaccines, passive immunization

## Abstract

The increasing incidence in systemic fungal infections in humans has increased focus for the development of fungal vaccines and use of monoclonal antibodies. Invasive mycoses are generally difficult to treat, as most occur in vulnerable individuals, with compromised innate and adaptive immune responses. Mortality rates in the setting of our current antifungal drugs remain excessively high. Moreover, systemic mycoses require prolonged durations of antifungal treatment and side effects frequently occur, particularly drug-induced liver and/or kidney injury. The use of monoclonal antibodies with or without concomitant administration of antifungal drugs emerges as a potentially efficient treatment modality to improve outcomes and reduce chemotherapy toxicities. In this review, we focus on the use of monoclonal antibodies with experimental evidence on the reduction of fungal burden and prolongation of survival in in vivo disease models. Presently, there are no licensed monoclonal antibodies for use in the treatment of systemic mycoses, although the potential of such a vaccine is very high as indicated by the substantial promising results from several experimental models.

## 1. Introduction

The increased numbers of immunocompromised hosts, global travel, climate alterations, and the common use of invasive devices have resulted in significant increases in rates of invasive mycoses. The medical mycology community has responded to this crisis by seeking out new approaches to combat these diseases, including through the development of vaccines and monoclonal antibodies (mAbs) [1,2,3,4,5] Recent studies show that systemic mycoses are a leading cause of morbidity and mortality in the US, being responsible for the deaths of more than 1.6 million people [2,6] with annual costs over $7.2 billion dollars [7,8,9]. There is, however, a significantly poorer awareness of the global incidence of systemic mycoses reinforced by the typical lack of transmissibility of fungal diseases [10,11]. There are currently ~120,000 identified fungal species [12], with only a few hundred capable of causing disease in humans. Of these, only a small number affect healthy people [13,14], differing in severity from mild to severe systemic infections [2,7].

Systemic mycoses in humans appear when there is poor host control of yeasts, hyphal fragments, spores or conidia, which predisposes the infection to progress through the bloodstream. Thereafter, the conidia/hyphal fragments or yeast cells can reach any organ [5,15]. Systemic mycoses are frequently difficult to treat, as most of them occur in vulnerable individuals, with defective innate and adaptive immune responses [2,14,16]. Diseases with worldwide occurrences, such as candidiasis and aspergillosis, are frequent and severe, and are currently causing the highest rate of hospitalization due to fungi [1,7,15]. Estimated costs of care for patients with only four of the major mycoses—aspergillosis, candidiasis, cryptococcosis, and histoplasmosis—have been calculated as up to $5.1 billion dollars annually in the US [7,17].

Even though treatment of human disease with antibiotics, immunosuppressive drugs, and anti-cancer medication have improved survival expectations in the setting of diverse diseases, these therapeutic interventions can also cause adverse effects leading to increased susceptibilities to microbial diseases, including viral, bacterial, parasitic, and fungal pathogens [2,15]. Systemic mycoses have been increasing globally, as a result of surgical interventions and therapeutic drugs as well as the ongoing HIV epidemics [2,5,14,18]. Individuals at high-risk of developing invasive of invasive fungal diseases include genetically immunocompromised persons, such as those with chronic or persistent autoimmune inflammatory diseases (e.g., rheumatic, dermatological, gastrointestinal, and neurological disorders), and individuals with hemato-oncological syndromes. Patients with acquired immune-related conditions, such as people undergoing immunomodulatory treatments, including cancer chemotherapy, organ/stem cell transplantation, corticosteroid or monoclonal antibody-induced immune suppression are likewise at high risk [2,19]. Invasive surgical interventions, intensive care (e.g., intubation and invasive catheters), and parenteral nutrition [1,2,20,21,22,23] heighten risk for invasive mycoses. Patients suffering antibiotic-induced alteration of the microbiota (gut dysbiosis) by the use of broad-spectrum antibiotics and/or antimycotics for prolonged periods are also susceptible to develop invasive fungal diseases [2,6]. Healthy individuals massively exposed to fungi (e.g., during construction, earthquakes, and tree cutting) as well as newborns and the elderly people are also included in the high risk category [1,2,20,21,22,23].

Recently, an unprecedented frequency of pathogenic fungi resistant to the limited, poorly available antifungal drugs has been reported [24]. Years of prolonged use of these drugs in many areas such as medical and veterinary clinics and agricultural sites have caused important modifications in the global microbiome, with the emergence of drug resistant fungal pathogens [15,24]. This resistance to antifungal drugs shown by many fungal species is mainly associated with immunocompromised individuals [9]. The toxicity of these medications is an important limitation to their use [4,10,24] and, in order to avert the universal failure in the management of fungal infections, new therapeutic strategies to treat systemic fungal infections are urgently needed [24,25].

Motivated by the recent advances in our understanding of host-fungus interactions, enriched by powerful molecular biological tools [1,15], there has been an exciting increased interest in mAbs as a possible alternative modality to treat mycoses, leading to a renewed focus on promising anti-fungal immunotherapies [1,16,26]. In this scenario, a focus on therapies based on mAbs against fungal infections [9], especially mycoses which do not respond to prophylactic vaccination [27], is urgently needed.

This review aims at updating the progress made in the field of therapeutic antifungal vaccines based on mAbs against systemic mycoses. It covers the current development of mAb vaccines and the contemporary challenges faced in this research field.

## 2. Immunotherapy Based on Monoclonal Antibodies

The role of antibody mediated immunity in fungal infections was elucidated as a result of the advances made ~40 years ago in hybridoma technology that allows for the generation of mAbs [25,26,28,29]. Today it is well known that invasive mycoses induce the production of diverse polyclonal antibody populations, which, depending on their specificity and isotype, may attenuate the effects of fungal infections; notably, they can be protective, non-protective or they can enhance the disease in the host [1,28,30]. B cells and antibodies have been reported to protect against infectious diseases by mitigating the host harm produced by the inflammatory response [29,31]. MAbs are highly specific and versatile molecules, since antibodies targeting a single epitope can be protective by promoting biological mechanisms such as complement-mediated lysis, stimulation of the pathogen phagocytic process by opsonization, cytokine release mediated by Fc or a direct antimicrobial effect [25,29,32]. MAbs can also alter biological functions of fungi, including modifying the release of extracellular virulence factors in vesicles [33,34].

As fungi can induce the production of protective antibodies, several studies have shown that these molecules can act as therapeutic vaccines against systemic mycoses. As a strategy to protect individuals who are unable to display a successful active immune response [35], passive antibody transference allows for the administration of protective mAbs against a specific pathogen, providing protection against infection in the absence of effective cellular immune mechanisms [1,25,28]. In Table 1, we summarize the advantages and disadvantages of using mAbs as a therapy for systemic mycoses.

MAbs-based immunotherapy in fungal diseases is supported by the vast antigenic differences between and among fungi and humans [25,29]. Native glycoproteins are promising targets for therapeutic antifungal vaccines, particularly cell wall glycoproteins [16,25]. Moreover, producing mAbs against intracellular targets, appears to also be an effective strategy to improve host defense [25,27]. Several antibodies against cell wall components have been described directed against displayed epitopes on *Candida albicans* and *Cryptococcus neoformans* [32,36]. Another well studied option for therapeutic mAb targets are heat shock proteins (Hsp), since they are conserved immuno- dominant antigens capable of eliciting cell mediated and humoral responses during infection [37,38,39]. By identifying surface molecules in fungi that interact with macrophage receptors, invasion processes in fungi capable to replicate within macrophages, such as *Paracoccidioides* spp., can be blocked by mAbs produced against these molecules [40].

To date, murine or human mAbs as well as genetically engineered antibody fragments have shown significant efficacy against fungi [25], including in immunocompromised animal models. In particular, many studies have reported that murine mAbs generated by hybridoma technology against fungi are protective in the murine model. Moreover, mAbs that are non-fungicidal could become fungicidal by labelling with a radiation emitter [1,41]. For their reduced toxicity, human antibodies are prized options in the search for immunotherapies to fungal infections [9]. Other advantages of using mAbs therapy, include the longevity of immune responses provided by IgG immunoglobulins—weeks to months—meaning that these antibodies can remain in a protective titer during prolonged periods of time—with considerable activity per mass of protein, as all the immunoglobulin molecules are specific for the chosen antigen [1,27]. Also, the production of therapeutic mAbs is potentially shorter than the time required for the development of prophylactic antifungal vaccines [29].

Currently, the majority of clinically utilized mAbs are chimeric, humanized or fully human IgG1, produced by hybridoma technology [9,42]. The production of therapeutic antibodies demands the handle of considerable cultures of mammalian cells and a subsequent purification process, guarantying good manufacturing practice (GMP) conditions, conducting to elevated production costs and restraining the extended use of mAbs [43]. The extent of their use in clinical practice is highlighted by the fact that the pharmaceutical market expects to accomplish $125 billion in sales of mAb therapeutics by 2020 [9,26].

However, mAb-based immunotherapeutics involve rigorous selection of the antibody characteristics, since specificity, affinity, and isotype define their effect on the host immune response [1,44]. Indeed, the same antibody specificity yet distinct isotype can turn a protective antibody into a non-protective or into an antibody that exacerbates the disease [1,30,45].

**Table 1 jof-06-00031-t001:** Advantages and disadvantages of using protective monoclonal antibodies for the treatment of systemic mycoses.

Advantages	Disadvantages
- MAbs avoid selection of drug-resistant fungal strains (highly specific) [1,29,30].	- MAbs are highly specific, therefore can only be used after a precise diagnosis of the agent [29,30].
- MAbs provide immediate immunity against systemic mycosis pathogens [1,29].	- MAbs efficacy may be quickly reduced as the infection progresses with time [29,46].
- MAbs can reduce antifungal drug treatment durations by enhancing their effectiveness (Synergistic effect) [1,29,30].	- MAbs are much more expensive to produce than antimycotic drugs [1,23,29,30].
- MAbs avoid toxicity risks because they are directed specifically to pathogen epitopes [1,27,29,47].	- MAbs are more difficult to store and administer than the conventional antifungal therapies [1,23,29,30].
- MAbs do not alter the microbiota [29].	
- MAbs can be originally raised against a wide range of molecular epitopes [23,29].	

There are currently no therapeutic vaccines licensed against fungal infections, for human or veterinary use [1,9,25,48]. Nevertheless, several studies consider mAbs as key options in a new age of antimicrobial treatment [27,29,49,50,51,52]. Here we summarize the research on immunotherapy development based on mAbs.

## 3. Monoclonal Antibody Therapy Approaches in Systemic Mycoses

### 3.1. Cryptococcus spp.

*Cryptococcus neoformans* and *Cryptococcus gattii* are encapsulated yeasts, and are the main etiological agents of cryptococcosis. *C. neoformans* has a worldwide distribution and is commonly associated with immunocompromised individuals. Infection by human immunodeficiency virus (HIV) is the main risk factor in cryptococcosis, but patients under treatment with immunosuppressive drugs and other immunocompromised hosts are also at risk [53]. *C. gattii* occurs in relatively immunocompetent individuals, but some other risk factors, such as preexisting conditions of heart or lung disease, may contribute to infection by *Cryptococcus* spp. [54,55,56,57].

Amphotericin B (AmB), 5-flucytosine (5FC), and fluconazole (FLZ) remain the primary choices for the treatment of cryptococcosis. As first-line therapy of cryptococcal meningitis or severe pulmonary cryptococcosis, the use of amphotericin B (AmB) in combination with 5-flucytosine (5-FC), followed by maintenance therapy with fluconazole, often for months, are recommended [58,59,60].

Although the role of antibodies in host defense against fungal infections was initially uncertain, work over the last three decades in diverse laboratories studying cryptococcosis as well as other systemic mycoses have demonstrated their powerful utility [25]. In order to understand antibody-mediated immunity in host defense against fungal infections, *C. neoformans* has been one of the most studied pathogens [61]. Although some studies have shown different results [62,63], the vast majority of in vitro efforts have suggested that antibodies to *C. neoformans*, mainly involving immunoglobulin G (IgG), effectively enhanced in vivo killing of the yeast [64,65]. In her seminal work, Dromer et al. [66] through the immunization of mice using *C. neoformans* capsular polysaccharide purified from *C. neoformans* serotype A, obtained a mAb to a capsular polysaccharide, which reacted with glucuronoxylomannan (GXM) and was called E1. Moreover, this study also effectively demonstrated that the mAb could modify the course of experimental murine cryptococcosis [66]. The capsule of *C. neoformans* is an important virulence factor that can subvert the defense mechanisms of the host [67,68]. One mechanism of action for mAb against cryptococcal polysaccharide is that these antibodies bind to *C. neoformans* polysaccharide and promote the clearance of the polysaccharide antigen from the serum of animals and humans resulting in an enhanced opsonization of the microorganisms, both in vitro and in vivo [69,70,71,72]. Additional mAbs to capsular polysaccharide have similarly been shown to increase the survival of infected mice and reduce the fungal burden in the tissues [73,74], but they also were found to increase the efficacy of AmB and FLZ against *C. neoformans* [69,72,75].

Murine IgG1 mAb to *C. neoformans* polysaccharide, known as 18B7, was studied in a clinical trial with human immunodeficiency virus-infected patients who had been successfully treated for cryptococcal meningitis. Aiming at determining the safety and maximum tolerated dose of mAb 18B7 in humans, this study found that 18B7 was safe and reduced serum GXM when used at high doses [76]. The development of this mAb, however, was hampered by funding issues and was interrupted.

Mycograb®, a recombinant scFv anti-HSP90 human antibody (produced against the chaperone from *Candida albicans*), showed efficacy against various species of fungi, but development of this immunoglobulin derivative also faced production difficulties (reviewed in [25]). For *C. neoformans*, Mycograb® was tested in a murine model using combination therapy with AmB, caspofungin (CAS) or FLZ, and the addition of the recombinant scFV antibody to antifungal drugs was more effectively than drug treatment alone [77].

In addition to GXM, melanin is another important virulence factor described in *C. neoformans* [78]. To study the production of melanin by *C. neoformans* during infection, two anti-melanin mAb were produced [79,80], and the administration of them prolonged the survival of mice infected with lethal inocula of *C. neoformans* and also reduced the fungal burden [80]. The authors suggested that the protection mechanism presented by these mAbs depended on mAb binding to melanized cells of *C. neoformans* in vivo, interfering with the growth and replication of the yeast.

Rodrigues et al. [81], identified a glucosylceramide, from lipid extracts of *C. neoformans* and demonstrated that antibodies purified against this molecule, mainly IgG1, bound to different strains and serological types of *C. neoformans*. When examined by confocal microscopy, the glucosylceramide accumulated mostly at the budding sites of dividing cells. In addition, when human antibodies, purified from sera of patients with active infection, were added to cultures of encapsulated and acapsular yeast, budding and cell growth were impaired. With these data, the authors suggest that mAb binding of glucosylceramide interfered with the cell wall synthesis and impaired cell division of *C. neoformans*.

MAb 2G8 is an antibody that binds to the poorly branched polysaccharide laminarin [82]. In studies on *C. neoformans*, this mAb 2G8 inhibited yeast growth of encapsulated and acapsular yeast. MAb 2G8 binds to the cell wall of *C. neoformans* and, in sub inhibitory concentrations, reduced the density of the capsule without affecting the production of enzymes. Encapsulated fungal cells were opsonized by the antibody and were efficiently phagocytosed. In addition, a single administration of mAb 2G8 resulted in the reduction of fungal load in the brains and livers of infected mice [83].

Other effects of anti-*C. neoformans* mAbs, such as interference with capsular polysaccharide release and biofilm formation have also been described [84,85]. Hence, the mechanisms of the primary protection exerted by mAbs against *C. neoformans* probably involves the alteration of inflammatory responses, enhancing fungal clearance and decreasing tissue damage (reviewed in [28]). In addition, specific antibodies effectively alter the interaction of yeast and macrophages [86]. Characterization of mAbs can reveal characteristics that explain their effectiveness. McClelland and Casadevall [87], demonstrated that mAbs that differ in epitope specificity and protective efficacy can also cause differences in gene expression.

The efficacy of mAbs against *C. neoformans* depends on the isotype and epitope specificity [74,88]. A study with the anti-capsular IgM mAbs 12A1 and 13F1, which are protective and nonprotective, respectively, and are derived from the same B cell, showed that IgM efficacy against *C. neoformans* depends on the route of infection, inoculum, and Ab dose, in addition to their capacity to promote opsonization and agglutination in vivo [89]. By using complement-deficient animals infected with *C. neoformans*, the ability of IgG isotypes to protect and increase the survival time of animals, demonstrated that IgG does not act via complement pathways [90]. According to a study performed with immunodeficient mice, T cells and the Th1 cytokine, IFN-γ, were found essential for IgG1 protection [91]. In another study, Th1 and Th2 cytokines were show as necessary for protection conferred by mAbs, since both Th1 and Th2 cytokines influenced the effect of antibodies with different isotypes (reviewed in [92]).

The results obtained through passive immunization with mAbs may depend on the immunological parameters of the host, such as the availability of T and B cells and the production of Th1 and Th2 cytokines (reviewed in [93]). Therefore, deficiencies in the host immune response associated with the inflammatory effect of the mAb are conditions that will predict whether the treatment will be protective (reviewed in [44]).

A Th1-driven cell-mediated response is essential for the control of infection by *C. neoformans*. According to work already carried out, the mAbs protection capacity can be associated to a cellular immune response, acting as a regulator. One example is the ability of GXM-specific mAbs to attenuate the suppression effects that this component has on the host immune response [44,68]. In this sense, the use of mAbs to treat infections by *C. neoformans* can modulate the host immune response, helping to control the infection as a complementary therapy.

### 3.2. Sporothrix spp.

*Sporothrix schenckii* has been considered the main etiological agent of sporotrichosis, but recent molecular studies have identified other species with phenotypic and genetic similarities that are now included in the *Sporothrix schenckii* complex [94,95,96]. Among the species included in this complex, *S. schenckii* sensu strictu, *S. brasiliensis*, *S. globosa*, *S. Mexicana*, and *S. luriei* are currently the species responsible for causing the majority of sporotrichosis (reviewed in [97,98]). The differences in the degrees of virulence and host immune responses to the species in the *Sporothrix schenckii* complex are directly associated with disease severity [99,100].

Sporotrichosis is a subcutaneous chronic fungal infection that is most prevalent in tropical and subtropical regions [101]. The disease typically occurs after traumatic skin inoculation of the conidia and hyphae, found in the remains of contaminated plants and soil (reviewed in [102]). Zoonotic transmission can also occur, since domestic cats are considered a source of transmission of this mycosis due to *S. brasiliensis* in humans [103,104,105]. Clinical manifestations of sporotrichosis may vary, depending on the degree of immunosuppression of the host [106]. In severe disease, antifungal therapy is based on the administration of itraconazole, terbinafine or AmB with itraconazole or terbinafine also being used in less severe cases [107]. As an alternative in mild disease, local heating and potassium iodide can be used for the treatment of lymphocutaneous/cutaneous infections (reviewed in [108]. Ketoconazole has been utilized in some cases of feline sporotrichosis [109].

Host protection against *S. schenckii* occurs mainly by T cell-mediated immunity [110], but humoral immune responses can help in the control of fungal infection [76,111]. Notably, antibodies have long been known to react with *Sporothrix* antigens. Early results on *S. schenckii* rhamnomannans and peptidorhamnomanns showed that rabbit and human antisera reacted with the α-l-Rha (1-2)-α-l-Rha epitope [112] and more specifically with *O*-linked α-l-Rha (1-4)-α-d-GlcA and α-l-Rha (1-4)-[α-l-Rha (1-2)]-α-d-GlcA [113].

In a study conducted by Nascimento and Almeida [114], the production of antibodies during experimental infection by *S. schenckii* in mice was evaluated. Through immunoblotting analysis, mice infected with the M-64 strain produced specific IgG1 and IgG3 antibodies against a 70-kDa fungal protein, indicating that antigens from *S. schenckii* induced a specific humoral response after 2 weeks of infection. The same results were obtained when comparing different infection routes. In a subsequent study, the authors produced the IgG1 mAb (P6E7) against the 70-kDa glycoprotein to clarify the effect of passive immunization on infected mice. The administration of mAb P6E7 reduced the number of CFUs in the spleen and liver of the infected mice. The results were significant in mice that were immunized before and during infection by *S. schenckii* and in mice that received P6E7 when the infection was established. In addition, increased levels of IFN-γ were observed in the animals receiving mAb P6E7, and other studies have indicated that IFN-γ production is directly associated with protection against sporotrichosis [115].

Toledo et al. [116] described that mAbs against specific glycosphingolipids are able to interfere with fungal growth and differentiation. They found that an IgG2a mAb, designated MEST-3, directed to *Paracoccidioides brasiliensis* glycolipid antigen Pb-2 (α-Man(1-3)-α-Man(1-2)-IPC), inhibited differentiation and colony formation in *S. schenckii*. When tested in cultures, it also inhibited the growth and differentiation of *Paracoccidioides brasiliensis* and *Histoplasma capsulatum*.

Franco et al. [117] studied the mechanisms of mAb P6E7 to the 70kd *Sporothrix* glycoprotein during phagocytosis and determined that the fungicidal activity of macrophages increased in presence of immune-inactivated serum or mAb P6E7. MAb P6E7 increased the production of TNF-α, IL-10 and IL-1β. Almeida et al. [118] further evaluated the efficacy of mAb P6E7 against two virulent isolates of *S. schenckii* (1099-18 and 15383) and two isolates of *S. brasiliensis* (5110 and 17943). The strains studied had different protein profiles and all the isolates caused chronic evolution of the disease in infected mice. Through Western blotting, only the exoantigen of virulent *S. brasiliensis* 5110 isolate, contained the gp70, unlike that described by Castro, et al. [119]. In addition, analysis of the organs of animals infected with the described strains, showed that mAb P6E7 reduced the fungal load in the animals, with a more significant result in the spleen of mice infected with *S. schenckii* strain 1099-18. In the liver, reduction of fungal load was observed only at the initial stage of infection, indicating that a higher dose of mAb P6E7 could promote more efficient protection. The authors also describe that a mixed response pattern of Th1 and Th2, was observed in this study. Contrary to what was found in the study conducted by Nascimento and Almeida [114], the response pattern in that study did not succeed in controlling the infection caused by virulent strains of *Sporothrix schenckii* complex.

### 3.3. Paracoccidioides spp.

Thermodimorphic fungi of the *Paracoccidioides* genus cause paracoccidioidomycosis (PCM), an endemic granulomatous mycosis of the sub-tropical regions of Latin America, widespread from south of Mexico to north of Argentina, with a high prevalence in South America [120,121,122,123]. PCM generally affects male workers in close contact with the soil in rural areas. Humid regions, moderately high pluviosity, mild temperatures, sites nearby rivers and forests or recently deforested zones, are ecological areas where *Paracoccidioides* spp. grow saprophytically [120,123,124]. Based on phylogenetic differences the predominantly studied species in the complex are *P. brasiliensis* and *Paracoccidiodes lutzii*, and these species are similar in their clinical manifestations and pathogenesis [37,125,126].

PCM occurs when airborne conidia reach the lung alveoli through the respiratory route, and, due to the shift to body temperature, transform into the yeast form, which can spread to other anatomic sites such as liver, adrenal glands, spleen, oral mucosae, and skin [123,127]. PCM presents two major clinical forms: (a) Acute/subacute or juvenile, characterized by fast progress and significant mortality rate; (b) chronic or adult PCM, which represents 90% of cases, and is characterized by gradual progress, typically occurring in male individuals older than 30 years [120,123,125,126,128].

*P. brasiliensis* and *P. lutzii* are susceptible to most of the systemic antimycotic medications as well as sulfamthoxazole/trimethoprim [125,126] and the treatment for PCM involves antifungal chemotherapy administered for prolonged periods, varying from patient to patient, depending on the chosen drug, the clinical manifestations and the progress of the disease [111,126,129]. Generally, extended periods of treatment are required—two or even more years—with a considerable chance of relapse [111,129]. Drug options include sulfonamide derivates, amphotericin B, azole derivates and terbinafine, but the most commonly used are itraconazole or sulfamethoxazole/trimethoprim for mild to moderate disease and amphotericin B for severe manifestations [126,129]. Hence, treatment of PCM requires extended antifungal administration regimens, which are associated with high public health costs, significant patient difficulties to complete the drug regime, and substantial risks for drug toxicities [30,129].

The primary effective mechanism to control experimental and human PCM is granuloma formation by activation of a Th1 type cellular immune response [37,111,129,130]. Naturally, activated macrophages perform a fundamental role in the resistance to *Paracoccidioides* spp. infection, leading to antimicrobial phagocytic activity [40,131]. For establishing an effective protection against PCM, both innate immune response and adaptive immunity are essential [37,45]. In the past two decades, several studies have focused on the therapeutic administration of mAbs, proving that antibody mediated immune response can promote fungal clearance and attenuate experimental disease [32,37,40,111,129].

The widely studied gp43 glycoprotein [132,133] is the main diagnostic antigen of *P. brasiliensis* and its major cell epitope was mapped to the internal peptide known as P10 (QTLIAIHTLAIRYAN), which elicits a Th1 type immune response, by an IFN-γ-dependent mechanism [45,111,128,129,134]. Clinical studies have shown that almost 100% of patients with *P. brasiliensis* infections have antibodies against gp43, suggesting that protective anti-gp43 antibodies in those patients could be at low concentration; however, they were likely at levels that were insufficient to control the disease [111,135]. Buissa Filho et al. [111] validated the effect of IgG2a and IgG2b mAbs against gp43 in experimental PCM by showing that specific mAbs could reduce fungal burdens and pulmonary inflammation in association with increased IFN-γ and IL-12 production. Although the mAb mAb 3E (IgG2b) was highly effective, other mAbs did not modify disease. In vitro results showed that mAb 3E was also capable of stimulating yeast phagocytosis and increasing NO production in MH-S, J774.16 and primary macrophages.

Mattos Grosso et al. [40] subsequently demonstrated that IgG1 mAbs (B7D6 and C5F11) to a 70 kDa glycoprotein (gp 70) of *P. brasiliensis* protected mice in an experimental PCM model. Efficacy was achieved when both mAbs were administered together as demonstrated by reduced fungal burdens and histopathology that showed reduced numbers of granulomas and yeast cells in pulmonary tissues. The authors suggested that macrophages had an important role in the elimination of the fungus since mAbs B7D6 and C5F11 bind to gp70 present on *P. brasiliensis* yeast surface to facilitate the killing of yeast by phagocytosis [40].

An apparently unrelated secreted protein of 75-kDa with phosphatase activity was also determined to be highly immunogenic [32]. BALB/c mice infected with *P. brasiliensis* that were treated with mAbs 1G6 (IgM) and 5E7C (IgG2a) against the 75-kDa protein had significantly reduced fungal burdens as well as a reduction in the number and size of pulmonary granulomas. In this study, the authors suggested the possibility that 1G6 and 5E7C mAbs are effective immunomodulators as they interact with effector cells through the Fc region to inhibit inflammatory responses [32].

Da Silva et al. [136] utilized anti-melanin polyclonal antibodies obtained by immunization of mice with melanin ghosts generated to study this antigens role as a virulence factor in *P. brasiliensis*. The authors demonstrated the inhibitory effect of melanin on the phagocytosis of melanized yeast. They noticed that J774.16 macrophage-like cells challenged with melanized yeast opsonized with anti-melanin antibodies had an increased phagocytic index, and significant altering ROS concentrations when compared to nonopsonized melanized yeast. They also performed an inhibition assay, associating the effect of an anti melanin antibody with carbohydrates (mannan and N-acetyl- glucosamine), that suppress the internalization of melanized yeast and showed the best inhibition effect on non-melanized yeast phagocytosis. This suggested that there are many cell receptors involved in the phagocytosis mechanism and that *P. brasiliensis* can employ melanin to protect the yeast cells from macrophage internalization, also reducing ROS release in these phagocytic cells [136].

MAbs 7B6 and 4E12 generated against heat shock protein 60 from *H. capsulatum* attenuate experimental PCM using a *P. lutzii* strain (Pb 01) [37]. The mAbs 7B6 (IgG2b) and 4E12 (IgG2a) effectively reduced the pulmonary fungal burden, which is in marked contrast with the infection enhancing effect of mAb 7B6 in experimental histoplasmosis. Histopathological analyses showed significantly reduced fungal burdens and reduced pulmonary damage with compact granuloma formation, suggestive of a protective Th-1 cellular response, confirmed by cytokine assays [37].

More recently, polyclonal antibodies to acidic glycosphingolipids (GSL) purified from *P. brasiliensis* have been studied [137]. Remarkably, administration of these polyclonal antibodies 30 days after intratracheal infection resulted in a significant therapeutic response, including the reduction in granuloma size and numbers, resulting in low tissue injury. Lung cytokines also showed a significant increment in IFN-γ, IL-4 and IL-12 suggesting a mixed Th1 and Th2 immune response. In vitro assays with IFN-γ activated peritoneal macrophages showed an improved phagocytic index and enhanced fungicidal activity when the target yeasts were opsonized with anti-acidic GSL polyclonal antibodies [137]. The authors suggest that therapeutic protocols based on antibodies could be a promising strategy for the treatment of established PCM.

### 3.4. Histoplasma spp.

*H. capsulatum* is a dimorphic fungus with a worldwide distribution, endemic in North America—with high prevalence in the soil from the midwestern and southern regions of the US [138] and areas within Latin America, and is associated with outbreaks in particular regions of Africa and Asia [139]. Despite the cosmopolitan characteristic, the exact frequency of the mycosis is still unclear as only few laboratories in endemic countries are prepared to effectively establish the diagnosis, particularly since histoplasmosis may simulate prevalent diseases such as cancer or tuberculosis [140,141].

Histoplasmosis is usually acquired by inhalation of fungal propagules present in contaminated soil or from excrement [142]. The infection is expressed in a clinical range from asymptomatic respiratory condition, a mild influenza-like illness, to a disseminated and life-threatening systemic disease, particularly in individuals with AIDS [143,144,145,146,147,148]. The determinants to the infection severity encompass the magnitude of exposure, the host immunological condition and the strain virulence [149,150].

Itraconazole, voriconazole, and amphotericin B are the drugs of choice for clinically significant disease. These potent therapeutics, however, fail to prevent mortality in about 10% of hospitalized patients. Additionally, the antifungal agents need to be administered for extended periods or even life-long in immunocompromised patients [151].

Generally, as discussed, there is a consensus that antibody can variably contribute along with phagocytic cells and T lymphocytes to enhance the host immune response in systemic mycoses [91,152,153,154,155,156]. The default infection control is based on activation of cellular immunity [157,158]. Additionally, in histoplasmosis, reactivation of previously controlled foci of infection can occur along with immunosuppression [159,160]. Polyclonal antibody preparations may have insufficient quantities of protective antibodies to modify disease progress or may even include inhibitory antibodies, as certain antigens in natural infection may not be immunogenic [49]. MAb preparations, in contrast, consist of one type of immunoglobulin with a defined specificity for the desired target and a single isotype [27]. MAbs against *H. capsulatum* have been generated, targeting cell wall antigens including melanin [161], histone H2B [162], Hsp60 [163], M antigen [164], and a 70-kDa protein [165].

Melanization of *H. capsulatum* conidia and yeast may affect inflammation and immunity in the course of histoplasmosis. Previous studies with other fungi such as *C. neoformans*, suggest that melanin or melanin-like compounds can hamper treatment of the infection. *H. capsulatum* conidia and yeast cells can synthesize the pigment in vitro and during mammalian infection. MAbs to pigmented *H. capsulatum* cells showed reactivity with cell surface melanin. It is clear that the influence and importance of this pigment in *H. capsulatum* should be further explored [161].

Protective mAbs directed to histone H2B efficiently increased phagocytosis and macrophage fungicidal activity in vitro. Passive administration of these mAbs in murine histoplasmosis models, reduced fungal burden, decreased inflammation and prolonged survival [54]. IgG mAbs were generated using recombinant HSP60 [163], an immunodominant antigen expressed on the surface of *H. capsulatum* yeast cells [166,167]. IgG2a and IgG1 isotypes were protective, inducing a strong Th1 response, according with cytokine analysis. The mAbs reduced fungal burden, decreased tissue damage and prolonged survival. In contrast, IgG2b was disease-enhancing [168].

The M antigen is a glycoprotein used for diagnosis of acute histoplasmosis, as it induces the first precipitins during disease [160]. The antigen was also found to be expressed in the cell surface and to play a catalase function. IgM and IgG2a mAbs to the M antigen can efficiently opsonize *H. capsulatum*, which enhances phagocytosis and promotes host cell fungicidal activity. Mice challenged with opsonized yeast survived a lethal *H. capsulatum* inoculum [164]. However, as demonstrated by the IgG2b mAb to Hsp60 [168], not all mAbs to *H. capsulatum* surface antigens are protective, which was similarly demonstrated with a non-protective IgG1 mAb to 70-kDa cell surface antigen [165]; however, because the IgG1 to the 70-kDA is highly specific for *H. capsulatum*, it is a candidate for use in serological diagnosis [169,170].

The findings with mAbs against *H. capsulatum* indicate that both isotype and the antigen target are determinants of a protective response. Future research is needed to determine the role on protection of the surface antigen expression as well as the characterization of mAbs influence in the disease outcome.

### 3.5. Candida spp.

Among the opportunistic fungal pathogens, *Candida* species are responsible for 50% to 70% of systemic fungal infections. *Candida albicans* remains the most commonly isolated species, being responsible for high morbidity and mortality rates worldwide [171,172]. *Candida albicans* is an extremely adaptable pathogen and dangerous for human health since it can resist adverse conditions in different in vivo conditions, including nutrient availability, temperature variation, osmolarity, pH and oxygen availability [18]. Moreover, it can also resist against antifungals and form mixed biofilms with other species [173,174].

The FLZ is the most commonly used drug for treatment because it is a low cost azole with little toxicity [172]. The development of azole resistance in *Candida* species, however, has been extensively reported [175]. In the developed world, many clinicians initiate antifungal therapy with an echinocandin until susceptibilities demonstrate that fluconazole is appropriate. In the severe candidiasis, the gold standard drug is AmB, a polyene with high toxicity and nephrotoxicity. Although liposomal formulations have been developed in an attempt to improve treatment efficacy and decrease AmB toxicity, the high costs of the formulations limit their use [176].

For development of new antifungal therapies, the cell wall is a leading target because it contains significant source of antigens and proteins essential for fungal growth, virulence and pathogenicity [177]. Cell wall proteins such as Als3p, Sap2p, Hsp90p, and Hry1p as well as polysaccharides such as β-glucan and mannans have been mAb targets in vaccine production [178].

Als3p adhesin plays a role in host colonization and is required for adhesion, invasion, biofilm formation, escape the host immune system, and iron acquisition [178]. An IgM mAb called C7 (mAb C7), is capable of reacting with an Als3p peptide epitope and was produced by immunization of BALB/c mice with a stress > 200 kDa mannoprotein present in the *C. albicans* cell wall [179]. In other studies, the mAb C7 also reacted with *C. albicans* enolase and cross-reacted with tumor cell nuclear pore protein (Nup88) and β-actin [180,181]. In addition, mAb C7 also reacts with with other fungi such as *Candida krusei*, *C. tropicalis*, *C. glabrata*, *C. dubliniensis*, *C. lusitaniae*, *Cryptococcus neoformans*, *Scedosporium prolificans*, and *Aspergillus fumigatus* [182]. The mAb C7 is capable of inhibiting Hep-2 cell adhesion and *C. albicans* germination and filamentation, besides showing direct fungicidal activity [182]. Given findings with the NDV-3A vaccine and the multi-drug resistant *C. auris*) [183], mAbs to Als3 may also be a powerful therapeutic to combat this emerging fungal pathogen.

Mycograb®, the above described recombinant scFv anti-HSP90 human antibody, showed efficacy and synergy when combined with AmB, FLZ, and CAS against *Candida* species. Due to the failure to authorize marketing by the CHMP (Committee for Medicinal Products for Human Use) a new formulation, called Mycograb C28Y variant, has been developed in recent years, in which some amino acids have been replaced. Unfortunately, in the first in vivo trials, the formulation proved to be inefficient in a murine candidiasis model [184,185].

MAbs against β-(1→3)-d-glucan, an essential component of the fungal cell wall, were produced and named mAb 5H5 (IgG3 class) and mAb 3G11 (IgG1 class). These mAbs reacted with yeast and filamentous fungi such as *Aspergillus*, *Candida*, *Penicillium*, and *Saccharomyces cerevisiae*. Both mAbs were able to inhibit *A. fumigatus* conidial germination during the first few hours, facilitate fungal phagocytosis by macrophages in situ and act in synergy with FLZ, decreasing drug concentration compared to *C. albicans* monotherapy in vitro. In addition, protective efficacy in mice vaccinated with a single injection followed by a lethal dose to *C. albicans* infection was obtained. In this case, a higher survival rate was observed in mice receiving mAb 5H5, which has high affinity and specificity as an IgG3, capable of inducing antibody-dependent cytotoxicity and cell phagocytosis more effectively than IgG1, being able to target mother cells, the main infective propagules [4].

The treatment time after infection is essential to achieve good antifungal immunotherapy results. The IgM mAb B6.1 specific for β-1, 2-mannotriose, was administered 1 h after murine infection with 5 × 105 *C. albicans* cells, reducing 28% CFU (colony forming unit), while mice treated 2 h after infection were not protected [186]. To improve treatment efficacy, the mAb B6.1 has been evaluated in combination therapy with AmB and FLZ for treatment of disseminated candidiasis [186,187]. MAb B6.1 in combination with AmB (0.5 mg/kg) administered 1 h after infection increased the survival time in mice equivalent to two doses of AmB at 2 mg/kg. Administration of MAb B6.1 and AmB after 2 h of infection helped to reduce disease severity [186]. Combination therapy of MAb B6.1 and FLZ (0.8 mg/kg) was also effective in increasing the survival rate of mice, equivalent to 3.2 mg/kg FLZ monotherapy administration [187]. Thus, the combination of mAb and antifungal drugs improves therapeutic efficacy by reducing the commonly required dose of chemotherapy and hence the side effects resulting from antifungal toxicity.

Several studies describe the protective effect of antibodies against candidiasis that represent strong prophylactic vaccine candidates. For example, mAb C7 and mAb 2G8 that react with Als3p protein [82,188], anti-iC3b receptor-specific mAb [189] and anti-whole cell mAb AB119, against recombinant *Candida albicans* Hyr1 cell wall protein was protective in a prophylactic mouse model of systemic *Candida* infection [9].

### 3.6. Aspergillus spp.

Aspergillosis is a clinically variable fungal infection, with presentations ranging from minor syndromes, such as bronchopulmonary allergic reactions, to severe chronic pulmonary involvement and disseminated, invasive disease [190,191]. The disease develops mainly in severely immunocompromised patients, suffering from hematologic malignancies, who underwent solid organ transplants (mainly lungs, kidney, or liver) or hematopoietic stem cell transplants as the main risk groups [192].

Aspergillus spp. are ubiquitously present in the environment, and are considered a common nosocomial contaminant [193,194,195] due to the dispersion of conidia, which, when inhaled, may cause IA in susceptible hosts [196]. Aspergillus fumigatus is the most common species followed by *A. flavus*, *A. niger*, *A. terreus*, and *A. nidulans* [190,191]. The correct identification of the infecting species is important due to the observed differences between the survival of infected patients and the different species involved [191]. *A. niger* infections are associated with best patient outcomes [192], whereas *A. terreus* infections are linked to the worst outcomes [197].

The differences in survival between the infecting species in IA are mainly attributed to differences in antifungal susceptibility within the genus [197,198]. The most commonly used antifungal agents against IA include azoles (particularly Voricanazole and Pozoconazole), echinocandins, and Amphotericin B, and these drugs may be given in combination [191,192,197,198]. Voriconazole is the frequently used first line agent [192], due to its efficacy and toxicity profiles [199]. Notably, *A. terreus* is predominantly resistant to amphotericin B [197]. Even though there are various drugs available for antifungal treatment of AI, fatality rates remain unacceptably high [192,200,201]. Hence, the development of new therapeutic approaches to this disease are increasingly necessary to face the growth of the immunocompromised population [202].

Immunity to aspergillosis is well described [203]. Although alveolar macrophages are an important first line of defense for killing Aspergillus spp. conidia, neutrophils, with the function of clearing the germinated hyphae and conidia that escape macrophage intracellular killing [204], are recognized as the main agents in the anti-Aspergillus innate immune response [203]. Adaptive cell-mediated immunity is outcome-defining, exerting a role in recruiting phagocytes and stimulating fungal intracellular killing, with Th1 and Th17 responses associated to protective responses [201,204]. Th2 responses are classified as non-protective [201]. Antibody-mediated responses remain poorly characterized in aspergillosis [202,204]. However, promising findings have been identified using mAbs to impair fungal adherence properties [205], prevent conidial germination [206,207] or directly kill fungal elements [208,209].

Several in vitro studies exploring the potential effect of mAbs raised against *Aspergillus* species have achieved promising results that could lead to their future use in treatment. Kumar and Shukla [205] used *A. fumigatus* secreted proteins to generate an IgM mAb, AK-14, that bound to an as yet undetermined carbohydrate-motif on fungal proteins. The mAb reacted with the cell surface of both *A. fumigatus* conidia and hyphae as well as to proteins from *A. flavus* and the dermatophyte *Trichophyton mentagrophytes*. Attachment tests showed that mAb AK-14 reduce the adhesion of *A. fumigatus* conidia to fibronectin in 70%, thus potentially reducing fungal pathogenicity.

Yadav and Shukla [210] produced an IgM mAb, R-5, that reacted with a 48 KDa protein present in conidia and hyphae of three *Aspergillus* species, *A. fumigatus*, *A. flavus*, and *A. niger*. The protein was identified as enolase, which is an important adherence factor for various mycopathogens, capable of binding to the host’s plasminogen [211]. MAb R-5 inhibited of spore germination by 88.3% in *A. fumigatus*, 57.4% in *A. flavus*, and 30.6% in *A. niger*. It inhibited growth by 24.1%, 13.3% and 8.8% in *A. fumigatus*, *A. flavus*, and *A. niger*, respectively. In a prophylactic murine disease model, R-5 reduced *A. fugmigatus* fungal burden by 85.9% and significantly increased survival [210].

Regarding studies that explored animal models of invasive aspergillosis to test the efficacy of mAbs, additional interesting results are found in the literature. Frosco et al. [199] were the first investigators to describe a mAb against *A. fumigatus*, using native and denatured elastase from the fungus, which is a secreted enzyme associated with the pathogenesis of lung diseases. Mice were immunized and hybridoma cell lines yielded five different antibodies against the enzyme with a considerable ability to inhibit the enzymatic activity, KD5 (IgG1), GD11 (IgG1), BB11 (IgG2a), MB8 (IgG2a), and CCIII 19 (IgG1) [212]. The application of a cocktail containing all five antibodies resulted in the survival of 16% of treated mice, when compared to 0% survival of the control groups, although no replicas of the experiment were explicitly done by the authors [213].

Chaturvedi et al. [206] described an IgG1 mAb, A9, that bound to a 95 KDa unidentified glycoprotein found in *A. fumigatus’* cell wall and the mAb efficiently labeled both hyphae and conidia of *A. fumigatus*. In vitro tests showed that the mAb inhibited *A. fumigatus* growth by 94.8%. The mAb was also effective against *A. flavus* and *A. terreus* (>50%), but it had limited inhibitory activity against *A. niger* (<20%). MAb9 also inhibited conidial germination of A. fumigatus by ~50%. Additionally, in vivo studies with this mAb showed promising results in a prophylactic disease model [206]. In 2008, Chaturvedi et al. [214], produced three hybridoma cell lines to a *A. fumigatus* cell-wall antigen extract. Of these, the IgM mAb-7 bound the fungal cell wall protein catalase B and the mAb inhibited *A. fumigatus* growth. In vitro experiments showed the reduction of fungal CFU by more than 76% and there was visual damage to 21.18% of hyphal population.

Wharton et al. [215] used a previously described antibody against streptococci oligosaccharides, to evaluate its binding capacity onto *Aspergillus* species. Immunofluorescence results revealed that the mAb SMB-19 would bind to *A. fumigatus’*, *A. flavus’*, and *A. niger’s* hyphae and conidia. Although the paper mainly explores the prophylactic activities of SMB-19, its administration simultaneously with the infection with *A. fumigatus* conidia in Ncf1m1J/J mice, which are highly susceptible to aspergillosis infection by inhalation, was tested. Passive immunization with SMB-19 increased mice survival, with 70% of the infected animals surviving until day 11, when all non-treated mice were dead, and 20% alive at day 21 [215].

An antibody from a patient with IA and its scFv fragment described by Schutte et al., named MS112-IIB1, was found using the Chitin Ring Formation (Crf) cell wall transglycosidase 2 (CRF-2) from *A. fumigatus*. The mAb bound CRF-2 at subnanomollar concentrations [207]. Chauvin et al. [207] explored its capacity to recognize other isoforms of Crf conserved in isolated *A. fumigatus*, encoded by the CRF1 gene. A high affinity for the isoforms was found, and the biologic activity of the scFv was further explored. A decrease of 23.6% in the growth of *A. fumigatus* mutant strain, capable of expressing various CRF1 isoforms, was found in vitro. The inhibition occurred only when the antibody was applied to 0-h germinating spores, and not after 6 and 24 h, indicating that the activity exclusively affected the resting or early germinating spores. Treatment of rats with MS112-IIB1 at the same time and 32 h after the infection with *A. fumigatus* conidia, showed no decrease in the fungal load, but the antibody fragment promoted a significant increase in the number of neutrophils and macrophages at the site of infection [207].

Polonelli et al. prepared an anti-idiotypic antibody using an antibody that neutralizes a killer-toxin (KT) against *Candida*. This KT binds to fungal cell receptors to exert its cytotoxicity, and as both the receptor and the anti-KT mAb are complementary to the KT structure, some anti-KT antibodies will have a sequential structure similar or mimetizing the receptor. If an anti-idiotypic antibody against the anti-KT-mAb is generated, its Fab structure might mimetize the structure of the KT, being able to bind the same receptor that the KT binds and then cause cell death. By using a mAb against the killer-toxin 4 (KT-4), that has activity against *C. albicans*, to immunize mice, Polonelli et al. have obtained the IgM K10 [216]. Cenci et al. tested the activity of mAb K10 against *A. fumigatus* and described the in vitro capacity of delaying spore germination. Studies with the use of a T-cell depleted bone marrow transplant model in mice, revealed that IgM K10 could protect all mice treated for more than 60 days, while the entire group of control mice succumbed to aspergillosis at day 6 [208].

The work of Appel et al. [195] used mAbs fused with the aliinase enzyme, capable of converting one inert substance extracted from garlic, aliin, into the bioactive fungicidal compound aliicin. The application of the fused protein with aliin only results in the conversion into aliicin in the proximity of the fungal surface, causing minimal damage to the host cells surrounding it. Whole-cell *A. fumigatus* conidia and hyphae were used as antigens to immunize mice, and the obtained splenocytes generated a hybridoma against the fungal surface. The IgM mAb MPS5.44 was obtained from one of these hybridomas with affinity for *A. fumigatus*, although the specific epitope that bound to the mAb was not characterized. The MPS5.44-Aliinase conjugate was found to exert dose-dependent fungicidal activity, with 100% fungal clearance at 10 nM, demonstrating a high activity in comparison with the unconjugated aliinase, both applied in cohort with aliin. In vivo tests using a pulmonary model of aspergillosis resulted in the survival of up to 85% of the MPS5.44-Aliinase conjugate treated mice.

### 3.7. Coccidioides spp.

The ethiologic agents of coccidioidomycosis, or valley fever are *Coccidioides immitis* and *C. posadasii* [217,218]. Coccidioides spp. are dimorphic pathogens, growing as hyphae in its saprobic phase and transforming into endosporulating spherules in its parasitic phase. When inhaled, arthroconidias, produced by asexual reproduction, cause a progressive primary pulmonary infection in healthy individuals. Inside the host, artroconidias transform into spherules, in which endospores grow, and once mature, by rupture of these spherules, endospores are released and spreaded into the body, often causing secondary infections, and repeating the infection cycle [217,218,219,220].

Over the years, efforts have been made to generate a successful vaccine for coccidioidomycosis. Previous studies reported that B cells are needed for a successful protective vaccine response in animal model [217,221,222] and could play an important role in the control of *Coccidiodes* spp. infection (reviewed in [217,220]). Currently, the role of B cells in a successful host response to *Coccidioides* spp. is not fully established and additional research in this field is needed [217]. Earlier researches demonstrated that high titers of complement fixing antibodies are a bad prognostic sign in human coccidioidomycosis (reviewed in [220]).

However, there is evidence suggesting that humoral immunity is not protective in coccidioidomycosis [220,221]. A recent review published by Donovan et al. [217] addresses about the role of B cells in the host response. Authors highlighted that there is no substantial evolution in the knowledge until now and maybe the use of monoclonal antibodies, as showed previously with other fungal infection, may be useful for describe protective antibodies. Although previous findings indicate the importance of B cells, authors reviewed data showing that C57BL/6 mice treated with anti-CD20 (expressed on B-Cells and may be involved in the regulation of B-Cell activation and proliferation) developed the same immunity than non-treated mice [217]. For the development of a vaccine to treat coccidioidomycosis, efforts should be focused on establishing a vaccine that elicit a protective cellular immune response [220].

### 3.8. Pneumocystis spp.

*Pneumocystis jirovecii* (previously described as *Pneumocystis carinii*) is an opportunistic fungal pathogen that affects mainly patients with impaired immunity, specially patients with HIV/AIDS [223,224,225]. Other major risk groups include patients with cancer, transplant recipients and patients taking diverse immunosuppressive therapies. Patients with *P. jirovecii* pneumonia (PjP) usually present with dyspnea, dry cough, normal temperatures low-grade fevers, and, in severe cases patients, may have respiratory failure. Asymptomatic lung colonization may be diagnosed in non-HIV/AIDS or non-immunosuppressed patients, who may become reservoirs for dissemination to immunocompromised patients since *Pneumocystis* can be transmitted from person to person by airborne route (reviewed in [226,227]). Furthermore, even when the infection resolves, *P. jirovecii* can persist colonization in the host for months [19].

Corticosteroids, typically used to suppress inflammation, do not significantly benefit PjP patients [228,229,230,231,232]. Moreover, patients undergoing immunomodulatory therapies with monoclonal antibodies, including TNF-alpha antagonists, CD52 antagonists, CD20 receptor blockers and many cytokine inhibitors have been associated with the development of *P. jirovecii* infections [19]. Research on a murine model demonstrated the role of B-cells perform in the immune response against PjP [233]. Anti-CD20-based chemotherapy is a conventional treatment for patients with hematological malignances who are HIV-uninfected, since it offers superior patient survival [234]. According with Elsegeiny et al. [235] CD20+ B-cells could be required to prime CD4+ T-cells against *Pneumocystis* spp. In this study mice treated with anti-CD20 had reduced cell type II responses to *Pneumocystis*, and CD4+ cells from depleted mice had an intrinsic impairment in their capacity to clear *Pneumocystis* cells, indicating that nosocomial PjP could be related to the patient being immunosuppressed and infected by an antigenically distinct strain of *Pneumocystis* that prior humoral immunity is ineffective at avoiding [235]. *Pneumocystis* spp. has a dynamic extracellular proteome resulting from changing major surface glycoproteins, which could be employed to resist the host immune response [235,236].

The first-line therapy for active *Pneumocystis* infection, for *Pneumocystis* prophylaxis [5,6,7], and the treatment of *Pneumocystis*-related IRIS is trimethoprim-sulfamethoxazole (TMP-SMX) [237,238]. However, TMP-SMX can be associated with several side effects, such as rash and cytopenia, and has the limitation of not being recommended for patients with renal insufficiency or sulfa-derivatives allergy [239,240,241], which can limit its use [242]. Other treatments indicated as second-line therapy, such as pentamidine and dapsone, tend to have higher failure rates [239,240].

Despite the availability of effective antibiotics, because of the failure to adequately control immunopathogenesis during the treatment, the mortality rates among patients developing severe PjP remain high [224]. Since the immune-mediated inflammatory lung injury is a major component of PjP pathogenesis [243,244,245], alternative strategies to suppress immunopathogenesis while eradicating the infection are badly needed.

The role of mAb therapy in controlling exacerbated inflammation related to infection has been poorly explored [225]. However, evidence suggests antibodies can provide protection against *P. jirovecii*. Passive transfer of serum elicited by immunization [246] or prophylaxis with monoclonal antibodies that recognize surface epitopes on *P. jirovecii* provided protective immunity in immunodeficient mice [35].

Sulfasalazine (SSZ), an immunomodulator, was found to reduce the severity of PjP in mice when administered before the onset of the pathogenic pulmonary immune response [247]. The combination of SSZ plus anti-*Pneumocystis* antibodies reduced the severity of PCP-related respiratory impairment, enhanced macrophage phagocytosis of *Pneumocystis* cells within the lungs, accelerating their clearance, and promoted recovery in BALB/c mice [248]. These findings indicate that appropriate immunomodulation has the potential to improve the outcome in patients suffering PjP. Compared to nonspecific immunosuppressive agents such as corticosteroids, mAb therapy presents the convenience of being pathogen-specific [28,30,225].

Using a therapeutic protocol, Hoy et al. [225] showed that passive administration of a pool of IgG and IgM mAb against *Pneumocystis* along with antimicrobial therapy with SSZ could limit the inflammation related to PjP in severe combined immune-deficient (SCID) mice. In this study, infected and treated mice gained weight, had improved pulmonary function and demonstrated a more robust inflammatory response compared to the control group that received irrelevant antibodies plus SSZ. They associated the increased phagocytosis, the initial macrophage recruitment, the reduction of neutrophil population late in the progression of infection, and a switch in the macrophage phenotype from M1 to M2 to the improved inflammatory response [225].

### 3.9. Blastomyces spp.

Blastomycosis is a systemic mycosis that usually affects dogs and humans [249]. Is caused by an endemic, thermal dimorphic fungi from the genus *Blastomyces* when contracted by inhalation of the conidia produced in its mold phase. Once that conidia reach the lungs, because of the temperature switch transforms into pathogenic yeasts, which can evade the host immune defenses and lead to pneumonia. In the yeast form, *Blastomyces* spp. is phagocytized by macrophages and other immune cells and can spread by blood circulation to other organs, producing a disseminated infection (reviewed in [250,251]). The most common form of *Blastomycosis* is the pulmonary disease and diverges from patient to patient from mild, self-limited infection to a severe respiratory distress syndrome. *Blastomyces dermatitidis*, *Blastomyces gilchristii*, and *Blastomyces helices* cause blastomycosis in North America, another species, *Blastomyces percursus* cause the mycosis in Africa. Itraconazole, voriconazole and posaconazole are recommended as first-line therapy in mild-to-moderate infection. As first-line therapy in severe cases patients are treated with amphotericin B (reviewed by [250,251]).

In a recent study Helal et al. [252] demonstrated by the first time the efficacy of a radiolabeled mAb (111In-400-2 mAb) to *Blastomyces dermatitidis*. The study demonstrated the capacity of an anti-(1-3)-beta-d-glucan mAb supplied with an alpha-emitter 213Bi to destroy *B. dermatitidis* cells both in vitro and in vivo. Moreover, the study demonstrated the efficacy of RIT targeting a fungal pan antigen. The results exhibited three times more accumulation of the 111In-400-2 mAb in the pulmonary tissue of the infected mice when compared to its accumulation in the non-infected mice 24 h after the mAb administration. The infected mice treated with the 400-2 mAb showed a reduction of 2 logs less CFU when compared to non-treated infected group [252].

## 4. Challenges and Perspectives of Therapeutic Antibodies to Fungal Infections

Antibody-based therapy has continuously evolved since the initial discovery of hybridoma technology. Most of the mAbs currently in the clinic are used for the treatment of cancer (38.9%), followed by autoimmune (25%), genetic (6.9%), infectious (5.5%), asthma (4.2%), cardiovascular (4.2%), hematologic (4.2%), and macular degeneration (2.8%) diseases. Other indications (8.3%) included transplant rejection (2.8%), bone loss, antidote, eczema and diabetes type 2 [253]. A recent survey of the number of mAbs in clinical studies showed that 575 mAbs are in Phase I/II and 70 mAbs in Phase III, and there are at least 61 in bispecific formats that are in clinical trials [254].

Hence, there are many factors that demonstrate the promise of mAb therapy for systemic mycoses. Antibodies mediate antimicrobial function through a variety of mechanisms, such as inhibition of microbial attachment, agglutination, neutralization, cytotoxicity, complement activation, and opsonization [1,20,29,46]. Given the current challenges of antifungal drugs in clinical use, immunomodulatory therapy, in synergy with our existing antifungal therapy, is an attractive option to enhance the immune system and clearance of the fungi.

The economic factor, though, is still not ideal for the large introduction of mAbs in the infectious diseases therapy due to the high costs of drug discovery and production. To date, only four mAbs were licensed by the US FDA for infectious disease therapy. Palivizumab was approved for use in the prevention of respiratory tract infection caused by Respiratory Syncytial Virus (RSV) [27,255]. Raxibacumab, in 2012, and Obiltoxaximab, in 2016, were approved for therapy in the treatment and prophylaxis of patients with exposure to or disease from *Bacillus anthracis* [27,256,257]. Bezlotoxumab was approved, in 2016, for treatment to prevent recurrence of *Clostridium difficile* infection [258]. Currently, there are no immunotherapeutics or vaccines approved for the treatment or prevention of fungal infections.

Despite difficulties, the growing resistance of microorganisms to the usual therapeutic drugs, the discovery of new and emergent pathogens, and our deepening knowledge into pathogen–host interactions, have increased the attention by the pharmaceutical industry, beyond the academic community, to the potential for therapeutic antibodies against mycoses. Although most mAb-based approaches are disease specific, there is also a potential of using mAbs to antigens presented on diverse fungi, such as melanin, HSP60 and other targets, to develop universal antifungal treatments, including through the use of mAbs to deliver lethal payloads to the fungi using radioimmunotherapy [41,259]. The decades of experience with mAbs in fungal diseases serves a strong foundation for increased efforts to bring mAb therapeutics against fungal pathogens into clinical practice.

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
