# Peer review of "Immunotherapy against Systemic Fungal Infections Based on Monoclonal Antibodies"

_jof, 2020, doi:10.3390/jof6010031_

Round 1
Reviewer 1 Report
This is a nice summary of the potential role of antibody for antibody in protecting the host against attack by fungal pathogens. There is really nothing significant to criticize with one exception. The role of antibody in protection against Pneumocystis was published by the Rochester group within a couple of months of Dr. Domer's, yet there is only a single mention of Pneumocystis. I would wager that at this point in time we know more about the role of antibody in interactions than with most other fungi. Is there a reason that Pneumocystis was ignored?
Reviewer 2 Report
Overall this is a very nice review that is easy to read and follow. However, in a way, it is not completely balanced, because it focuses exclusively on the positive effects of Mabs, but completely ignores challenges associated with Mab therapy. Eg: Limited improvement effect in many fungal diseases or no improvement effect, possibilities of toxic effects, high costs of antibody therapy, possible development of anti-antibodies, etc. This review would be much more thought-provoking if problems and their possible solutions would be discussed as well.
Besides this, there are only a few minor suggestions for corrections:
Ln 58-68: The risk factor paragraph could be strengthened. Perhaps it would be good to refer to contaminated indwelling catheters as a frequent source of blood-stream infections and immunosuppressive therapies in people with autoimmune diseases, etc.
Ln 95 I think "attenuating" was meant to be "promoting", since antibodies are protective when they promote these processes.
Ln138-211. Because anticryptococcal defenses rely vastly on the cell-mediated immunity, Ab therapies most likely could serve as a complementary therapy. This should be probably clarified/discussed somewhere in part 3.1.
